# Estimation Bias in Multi-Armed Bandit Algorithms for Search Advertising

**Min Xu**
Machine Learning Department
Carnegie Mellon University
minx@cs.cmu.edu

**Tao Qin**
Microsoft Research Asia
taoqin@microsoft.com

**Tie-Yan Liu**
Microsoft Research Asia
tie-yan.liu@microsoft.com

## Abstract

In search advertising, the search engine needs to select the most profitable advertisements to display, which can be formulated as an instance of online learning with partial feedback, also known as the stochastic multi-armed bandit (MAB) problem. In this paper, we show that the naive application of MAB algorithms to search advertising for advertisement selection will produce sample selection bias that harms the search engine by decreasing expected revenue and "estimation of the largest mean" (ELM) bias that harms the advertisers by increasing game-theoretic player-regret. We then propose simple bias-correction methods with benefits to both the search engine and the advertisers.

## 1 Introduction

Search advertising, also known as sponsored search, has been formulated as a multi-armed bandit (MAB) problem [11], in which the search engine needs to choose one ad from a pool of candidate to maximize some objective (e.g., its revenue). To select the best ad from the pool, one needs to know the quality of each ad, which is usually measured by the probability that a random user will click on the ad. Stochastic MAB algorithms provide an attractive way to select the high quality ads, and the regret guarantee on MAB algorithms ensures that we do not display the low quality ads too many times.

When applied to search advertising, a MAB algorithm needs to not only identify the best ad (suppose there is only one ad slot for simplicity) but also accurately learn the click probabilities of the top two ads, which will be used by the search engine to charge a fair fee to the winner advertiser according to the generalized second price auction mechanism [6]. If the probabilities are estimated poorly, the search engine may charge too low a payment to the advertisers and lose revenue, or it may charge too high a payment which would encourage the advertisers to engage in strategic behavior. However, most existing MAB algorithms only focus on the identification of the best arm; if naively applied to search advertising, there is no guarantee to get an accurate estimation for the click probabilities of the top two ads.

Thus, search advertising, with its special model and goals, merits specialized algorithmic design and analysis while using MAB algorithms. Our work is a step in this direction. We show in particular that naive ways of combining click probability estimation and MAB algorithms lead to sample selection bias that harms the search engine's revenue. We present a simple modification to MAB algorithms that eliminates such a bias and provably achieves almost the revenue as if an oracle gives us the actual click probabilities. We also analyze the game theoretic notion of incentive compatibility (IC)

and show that low regret MAB algorithms may have worse IC property than high regret uniform exploration algorithms and that a trade-off may be required.

## 2   Setting

Each time an user visits a webpage, which we call *an impression*, the search engine runs a *generalized second price (SP) auction* [6] to determine which ads to show to the user and how much to charge advertisers if their ads are clicked. We will in this paper suppose that we have only one ad slot in which we can display one ad. The multiple slot setting is more realistic but also much more complicated to analyze; we leave the extension as future work. In the single slot case, generalized SP auction becomes simply the well known second price auction, which we describe below.

Assume there are $n$ ads. Let $b_k$ denote the bid of advertiser $k$ (or the ad $k$), which is the maximum amount of money advertiser $k$ is willing to pay for a click, and $\rho_k$ denote the click-through-rate (CTR) of ad $k$, which is the probability a random user will click on it. SP auction ranks ads according to the products of the ad CTRs and bids. Assume that advertisers are numbered by the decreasing order of $b_i\rho_i$: $b_1\rho_1 > b_2\rho_2 > \cdots > b_n\rho_n$. Then advertiser 1 wins the ad slot, and he/she need to pay $b_2\rho_2/\rho_1$ for each click on his/her ad. This payment formula is chosen to satisfy the game theoretic notion of incentive compatibility (see Chapter 9 of [10] for a good introduction). Therefore, the per-impress expected revenue of SP auction is $b_2\rho_2$.

### 2.1   A Two-Stage Framework

Since the CTRs are unknown to both advertisers and the search engine, the search engine needs to estimate them through some learning process. We adopt the same two-stage framework as in [12, 2], which is composed by a CTR learning stage lasting for the first $T$ impressions and followed by a SP auction stage lasting for the second $T_{end} - T$ impressions.

1. Advertisers $1, ..., n$ submit bids $b_1, ..., b_n$.

2. **CTR learning stage**:
   For each impression $t = 1, ..., T$, display ad $k_t \in \{1, ..., n\}$ using MAB algorithm $\mathcal{M}$. Estimate $\widehat{\rho}_i$ based on the click records from previous stage.

3. **SP auction stage**:
   For $t = T + 1, ..., T_{\text{end}}$, we run SP auction using estimators $\widehat{\rho}_i$: display ad that maximizes $b_k\widehat{\rho}_k$ and charge $\frac{b_{(2)}\widehat{\rho}_{(2)}}{\widehat{\rho}_{(1)}}$. Here we use $(s)$ to indicate the ad with the $s$-th largest score $b_i\widehat{\rho}_i$.

One can see that in this framework, the estimators $\widehat{\rho}_i$'s are computed at the end of the first stage and keep unchanged in the second stage. Recent works [2] suggested one could also run the MAB algorithm and keep updating the estimators until $T_{\text{end}}$. However, it is hard to compute a fair payment when we display ads based using a MAB algorithm rather than the SP auction, and a randomized payment is proposed in [2]. Their scheme, though theoretically interesting, is impractical because it is difficult for advertisers to accept a randomized payment rule. We thus adhere to the above framework and do not update $\widehat{\rho}_i$'s in the second stage.

It is important to note that in search advertising, we measure the quality of CTR estimators not by mean-squared error but by criteria important to advertising. One criterion is to the per-impression expected revenue (defined below) in rounds $T + 1, ..., T_{\text{end}}$. Two types of estimation errors can harm the expected revenue: (1) the ranking may be incorrect, i.e. $\arg\max_k b_k\widehat{\rho}_k \neq \arg\max b_k\rho_k$, and (2) the estimators may be biased. Another criterion is incentive compatibility, which is a more complicated concept and we defer its definition and discussion to Section 4. We do not analyze the revenue and incentive compatibility properties of the first CTR learning stage because of its complexity and brief duration; we assume that $T_{\text{end}} >> T$.

**Definition 2.1.** Let $(1) := \arg\max_k b_k\widehat{\rho}_k$, $(2) := \arg\max_{k \neq (1)} b_k\widehat{\rho}_k$. We define the per-impression *empirical revenue* as $\widehat{\text{rev}} := \rho_{(1)}\frac{b_{(2)}\widehat{\rho}_{(2)}}{\widehat{\rho}_{(1)}}$ and the per-impression *expected revenue* as $\mathbb{E}[\widehat{\text{rev}}]$ where the expectation is taken over the CTR estimators. We define then the per-impression *expected revenue loss* as $b_2\rho_2 - \mathbb{E}[\widehat{\text{rev}}]$, where $b_2\rho_2$ is the oracle revenue we obtain if we know the true click probabilities.

**Choice of Estimator**  We will analyze the most straightforward estimator $\widehat{\rho}_k = \frac{C_k}{T_k}$ where $T_k$ is the number of impression allocated to ad $k$ in the CTR learning stage and $C_k$ is the number of clicks received by ad $k$ in the CTR learning stage. This estimator is in fact biased and we will later propose simple improvements.

## 2.2   Characterizing MAB Algorithms

We analyze two general classes of MAB algorithms: uniform and adaptive. Because there are many specific algorithms for each class, we give our formal definitions by characterizing $T_k$, the number of impressions assigned to each advertiser $k$ at the end of the CTR learning stage.

**Definition 2.2.**  We say that the learning algorithm $\mathcal{M}$ is *uniform* if, for some constant $0 < c < 1$, for all $k$, all bid vector $\mathbf{b}$, with probability at least $1 - O\left(\frac{n}{T}\right)$:

$$T_k \geq \frac{c}{n}T.$$

We next describe adaptive algorithm which has low regret because it stops allocating impressions to ad $k$ if it is certain that $b_k\rho_k < \max_{k'} b_{k'}\rho_{k'}$.

**Definition 2.3.**  Let $\mathbf{b}$ be a bid vector. We say that a MAB algorithm is *adaptive* with respect to $\mathbf{b}$, if, with probability at least $1 - O\left(\frac{n}{T}\right)$, we have that:

$$T_1 \geq cT_{max} \quad \text{and} \quad \left(c'\frac{b_k^2}{\Delta_k^2}\ln T\right) \geq T_k \geq \min\left(cT_{max}, \frac{4b_k^2}{\Delta_k^2}\ln T\right) \quad \text{for all } k \neq 1$$

where $\Delta_k = b_1\rho_1 - b_k\rho_k$ and $c < 1, c'$ are positive constants and $T_{max} = \max_k T_k$. For simplicity, we assume that $c$ here is the same as $c$ in Definition 2.2, we can take the minimum of the two if they are different.

Both the uniform algorithms and the adaptive algorithms have been used in the search advertising auctions [5, 7, 12, 2, 8]. UCB (Uniform Confidence Bound) is a simple example of an adaptive algorithm.

**Example 2.1.  UCB Algorithm**. The UCB algorithm, at round $t$, allocate the impression to the ad with the largest score, which is defined as $s_{k,t} \equiv b_k\widehat{\rho}_{k,t} + \gamma b_k\sqrt{\frac{1}{T_k(t)}\log T}$.

where $T_k(t)$ is the number of impressions ad $k$ has received before round $t$ and $\widehat{\rho}_{k,t}$ is the number of clicks divided by $T_k(t)$ in the history log before round $t$. $\gamma$ is a tuning parameter that trades off exploration and exploitation; the larger $\gamma$ is, the more UCB resembles uniform algorithms. Some version of UCB algorithm uses $\log t$ instead of $\log T$ in the score; this difference is unimportant and we use the latter form to simplify the proof.

Under the UCB algorithm, it is well known that the $T_k$'s satisfy the upper bounds in Definition 2.3. That the $T_k$'s also satisfy the lower bounds is not obvious and has not been previously proved. Previous analyses of UCB, whose goal is to show low regret, do not need any lower bounds on $T_k$'s; our analysis does require a lower bound because we need to control the accuracy of the estimator $\widehat{\rho}_k$. The following theorem is, to the best of our knowledge, a novel result.

**Theorem 2.1.**  *Suppose we run the UCB algorithm with $\gamma \geq 4$, then the $T_k$'s satisfy the bounds described in Definition 2.3.*

The UCB algorithm in practice satisfy the lower bounds even with a smaller $\gamma$. We refer the readers to Theorem 5.1 and Theorem 5.2 of Section 5.1 of the appendix for the proof.

As described in Section 2.1, we form estimators $\widehat{\rho}_k$ by dividing the number of clicks by the number of impressions $T_k$. The estimator $\widehat{\rho}_k$ is not an average of $T_k$ i.i.d Bernoulli random variables because the size $T_k$ is correlated with $\widehat{\rho}_k$. This is known as the sample selection bias.

**Definition 2.4.**  We define the *sample selection bias* as $\mathbb{E}[\widehat{\rho}_k] - \rho_k$.

We can still make the following concentration of measure statements about $\widehat{\rho}_k$, for which we give a standard proof in Section 5.1 of the appendix.

**Lemma 2.1.** *For any MAB learning algorithm, with probability at least $1 - O(\frac{n}{T})$, for all $t = 1, ..., T$, for all $k = 1, ..., n$, the confidence bound holds.*

$$\rho_k - \sqrt{(1/T_k(t))\log T} \leq \widehat{\rho}_{k,t} \leq \rho_k + \sqrt{(1/T_k(t))\log T}$$

### 2.3 Related Work

As mentioned before, how to design incentive compatible payment rules when using MAB algorithms to select the best ads has been studied in [2] and [5]. However, their randomized payment scheme is very different from the current industry standard and is somewhat impractical. The idea of using MAB algorithms to simultaneously select ads and estimate click probabilities has proposed in [11], [8] and [13] . However, they either do not analyze estimation quality or do not analyze it beyond a concentration of measure deviation bound. Our work in contrast shows that it is in fact the *estimation bias* that is important in the game theoretic setting. [9] studies the effect of CTR learning on incentive compatibility from the perspective of an advertiser with imperfect information.

This work is only the first step towards understanding the effect of estimation bias in MAB algorithms for search advertising auctions, and we only focus on a relative simplified setting with only a single ad slot and without budget constraints, which is already difficult to analyze. We leave the extensions to multiple ad slots and with budget constraints as future work.

## 3 Revenue and Sample Selection Bias

In this section, we analyze the impact of a MAB algorithm on the search engine's revenue. We show that the direct plug-in of the estimators from a MAB algorithm (either unform or adaptive) will cause the *sample selection bias* and damage the search engine's revenue; we then propose a simple de-bias method which can ensure the revenue guarantee. Throughout the section, we fix a bid vector $\mathbf{b}$. We define the notations $(1), (2)$ as $(1) := \arg\max_k \widehat{\rho}_k b_k$ and $(2) := \arg\max_{k\neq(1)} \widehat{\rho}_k b_k$.

Before we present our main result, we pause to give some intuition about sample selection bias. Assume $b_1\rho_1 \geq b_2\rho_2... \geq b_n\rho_n$ and suppose we use the UCB algorithm in the learning stage. If $\widehat{\rho}_k > \rho_k$, then the UCB algorithm will select $k$ more often and thus acquire more click data to gradually correct the overestimation. If $\widehat{\rho}_k < \rho_k$ however, the UCB algorithm will select $k$ less often and the underestimation persists. Therefore, $\mathbb{E}[\rho_k] < \rho_k$.

### 3.1 Revenue Analysis

The following theorem is the main result of this section, which shows that the bias of the CTR estimators can critically affect the search engine's revenue.

**Theorem 3.1.** *Let $T_0 := \frac{4n}{\rho_1^2}\log T$, $T_{min}^{adpt} := 5c'\left(\sum_{k\neq1}\frac{\max(b_1^2, b_k^2)}{\Delta_k^2}\right)\log T$, and $T_{min}^{unif} := 4\frac{nb_{max}^2}{c\Delta_2^2}\log T$. Let c be the constant introduced in Definition 2.3 and 2.2.*

*If $T \geq T_0$, then, for either adaptive or uniform algorithms,*

$$b_2\rho_2 - \mathbb{E}[\widehat{rev}] \leq \left(\left(b_2\rho_2 - b_2\mathbb{E}[\widehat{\rho}_2]\frac{\rho_1}{\mathbb{E}[\widehat{\rho}_1]}\right) - O\left(\sqrt{\frac{n}{T}\log T}\right) - O\left(\frac{n}{T}\right)\right).$$

*If we use adaptive algorithms and $T \geq T_{min}^{adpt}$ or if we use uniform algorithms and $T \geq T_{min}^{unif}$, then*

$$b_2\rho_2 - \mathbb{E}[\widehat{rev}] \leq \left(\left(b_2\rho_2 - b_2\mathbb{E}[\widehat{\rho}_2]\frac{\rho_1}{\mathbb{E}[\widehat{\rho}_1]}\right) - O\left(\frac{n}{T}\right)\right)$$

We leave the full proof to Section 5.2 of the appendix and provide a quick sketch here. In the first case where $T$ is smaller than thresholds $T_{min}^{adpt}$ or $T_{min}^{unif}$, the probability of incorrect ranking, that is, incorrectly identifying the best ad, is high and we can only use concentration of measure bounds to control the revenue loss. In the second case, we show that we can almost always identify the best ad and therefore, the $\sqrt{\frac{n}{T}\log T}$ error term disappears.

The $(b_2\rho_2 - b_2\mathbb{E}[\widehat{\rho}_2]\frac{\rho_1}{\mathbb{E}[\widehat{\rho}_1]})$ term in the theorem is in general positive because of sample selection bias. With bias, the best bound we can get on the expectation $\mathbb{E}[\widehat{\rho}_2]$ is that $|\mathbb{E}[\widehat{\rho}_2] - \rho_2| \leq O\left(\sqrt{\frac{1}{T_2}\log T}\right)$, which is through the concentration inequality (Lemma 2.1).

**Remark 3.1.** With adaptive learning, $T_1$ is at least the order of $O(\frac{n}{T})$ and $\frac{1}{T_2}\log T \geq \frac{\Delta_2^2}{c'b_2^2}$. Therefore, $\frac{\rho_1}{\mathbb{E}[\widehat{\rho}_1]}$ is at most on the order of $1 + \sqrt{\frac{n}{T}\log T}$ and $b_2\rho_2 - b_2\mathbb{E}[\widehat{\rho}_2]$ is on the order of $O(\Delta)$.

Combining these derivations, we get that $b_2\rho_2 - \mathbb{E}[\widehat{\text{rev}}] \leq O(\Delta_2) + O\left(\frac{n}{T}\right)$. This bound suggests that the revenue loss does not converge to $0$ as $T$ increases. Simulations in Section 5 show that our bound is in fact tight: the expected revenue loss for adaptive learning, in presence of sample selection bias, can be large and persistent.

For many common uniform learning algorithms (uniformly random selection for instance) sample selection bias does not exist and so the expected revenue loss is smaller. This seems to suggest that, because of sample selection bias, adaptive algorithms are, from a revenue optimization perspective, inferior. The picture is switched however if we use a debiasing technique such as the one we propose in section 3.2. When sample selection bias is 0, adaptive algorithms yield better revenue because it is able to correctly identify the best advertisement with fewer rounds. We make this discussion concrete with the following results in which we assume a *post-learning unbiasedness condition*.

**Definition 3.1.** We say that the *post-learning unbiasedness condition* holds if for all $k$, $\mathbb{E}[\widehat{\rho}_k] = \rho_k$.

This condition does not hold in general, but we provide a simple debiasing procedure in Section 3.2 to ensure that it always does. The following Corollary follows immediately from Theorem 3.1 with an application of Jensen's inequality.

**Corollary 3.1.** *Suppose the* post-learning unbiasedness condition *holds. Let* $T_0 \leq T_{min}^{adpt} \leq T_{min}^{unif}$ *be defined as in Theorem 3.1.*

*If we use either adaptive or uniform algorithms and $T \geq T_0$, then $b_2\rho_2 - \mathbb{E}[\widehat{\text{rev}}] \leq O\left(\sqrt{\frac{n}{T}\log T}\right)$.*

*If we use adaptive algorithm and $T \geq T_{min}^{adpt}$ or if we use uniform algorithm and $T \geq T_{min}^{unif}$, then*

$$b_2\rho_2 - \mathbb{E}[\widehat{\text{rev}}] \leq O\left(\frac{n}{T}\right)$$

The revenue loss guarantee is much stronger with the unbiasedness, which we confirm in our simulations in Section 5.

Corollary 3.1 also shows that the revenue loss drops sharply from $\sqrt{\frac{n}{T}\log T}$ to $\frac{n}{T}$ once $T$ is larger than some threshold. Intuitively, this behavior exists because the probability of incorrect ranking becomes negligibly small when $T$ is larger than the threshold. Because the adaptive learning threshold $T_{min}^{adpt}$ is always smaller and often much smaller than the uniform learning threshold $T_{min}^{unif}$, Corollary 3.1 shows that adaptive learning can guarantee much lower revenue loss when $T$ is between $T_{min}^{adpt}$ and $T_{min}^{unif}$. It is in fact the same adaptiveness that leads to low regret that also leads to the strong revenue loss guarantees for adaptive learning algorithms.

### 3.2 Sample Selection Debiasing

Given a MAB algorithm, one simple meta-algorithm to produce an unbiased estimator where the $T_k$'s still satisfy Definition 2.3 and 2.2 is to maintain "held-out" click history logs. Instead of keeping one history log for each advertisement, we will keep two; if the original algorithm allocates one impression to advertiser $k$, we will actually allocate two impressions at a time and record the click result of one of the impressions in the first history log and the click result of the other in the heldout history log.

When the MAB algorithm requires estimators $\widehat{\rho}_k$'s or click data to make an allocation, we will allow it access only to the first history log. The estimator learned from the first history log is biased by the selection procedure but the heldout history log, since it does not influence the ad selection, can be used to output an unbiased estimator of each advertisement's click probability at the end of the exploration stage. Although this scheme doubles the learning length, sample selection debiasing can significantly improve the guarantee on expected revenue as shown in both theory and simulations.

# 4 Advertisers' Utilities and ELM Bias

In this section, we analyze the impact of a MAB algorithm on advertisers' utilities. The key result of this section is the adaptive algorithms can exacerbate the "estimation of the largest mean" (ELM) bias, which arises because expectation of the maximum is larger than the maximum of the expectation. This ELM bias will damage advertisers' utilities because of overcharging.

We will assume that the reader is familiar with the concept of incentive compatbility and give only a brief review. We suppose that there exists a true value $v_i$, which exactly measures how much a click is worth to advertiser $i$. The utility per impression of advertiser $i$ in the auction is then $\rho_i(v_i - p_i)$ if the ad $i$ is displayed where $p_i$ is the per-click payment charged by the search engine charges. An auction mechanism is called *incentive compatible* if the advertisers maximize their own utility by truthfully bidding: $b_i = v_i$. For auctions that are close but not fully incentive compatible, we also define player-regret as the utility lost by advertiser $i$ in truthfully bidding $v_i$ rather than a bid that optimizes utility.

## 4.1 Player-Regret Analysis

We define $\mathbf{v} = (v_1, ..., v_n)$ to be the *true per-click values* of the advertisers. We will for simplicity assume that the post-learning unbiasedness condition (Definition 3.1) holds for all our results in this section. We introduce some formal definitions before we begin our analysis. For a fixed vector of competing bids $b_{-k}$, we define the *player utility* as $u_k(b_k) \equiv \mathbf{I}_{b_k \widehat{\rho}_k(b_k) \geq b_{k'} \widehat{\rho}_{k'}(b_k) \forall k'} \left( v_k \rho_k - \frac{\max_{k' \neq k} b_{k'} \widehat{\rho}_{k'}(b_k)}{\widehat{\rho}_k(b_k)} \rho_k \right)$, where $\mathbf{I}_{b_k \widehat{\rho}_k(b_k) \geq b_{k'} \widehat{\rho}_{k'}(b_k) \forall k'}$ is a 0/1 function indicating whether the impression is allocated to ad $k$. We define the *player-regret*, with respect to a bid vector $\mathbf{b}$, as the player's optimal gain in utility through false bidding $\sup_b \mathbb{E}[u_k(b_k)] - \mathbb{E}[u_k(v_k)]$. It is important to note that we are hiding $u_k(b_k)$'s and $\widehat{\rho}_k(b_k)$'s dependency on the competing bids $b_{-k}$ in our notation. Without loss of generality, we consider the utility of player 1. We fix $b_{-1}$ and we define $k^* \equiv \arg\max_{k \neq 1} b_k \rho_k$. We divide our analysis into cases, which cover the different possible settings of $v_1$ and competing bid $b_{-1}$.

**Theorem 4.1.** *The following holds for both uniform and adaptive algorithms.*

*Suppose $b_{k^*} \rho_{k^*} - v_1 \rho_1 \geq \omega(\sqrt{\frac{n}{T} \log T})$, then, $\sup_{b_1} \mathbb{E}[u_1(b_1)] - \mathbb{E}[u_1(v_1)] \leq O\left(\frac{n}{T}\right)$. Suppose $|v_1 \rho_1 - b_{k^*} \rho_{k^*}| \leq O(\sqrt{\frac{n}{T} \log T})$, then $\sup_{b_1} \mathbb{E}[u_1(b_1)] - \mathbb{E}[u_1(v_1)] \leq O\left(\sqrt{\frac{n}{T} \log T}\right)$.*

Theorem 4.1 shows that when $v_1 \rho_1$ is not much larger than $b_{k^*} \rho_{k^*}$, the player-regret is not too large. The next Theorem shows that when $v_1 \rho_1$ is much larger than $b_{k^*} \rho_{k^*}$ however, the player-regret can be large.

**Theorem 4.2.** *Suppose $v_1 \rho_1 - b_{k^*} \rho_{k^*} \geq \omega\left(\sqrt{\frac{n}{T} \log T}\right)$, then, for both uniform and adaptive algorithms:*

$$\forall b_1, \mathbb{E}[u_1(b_1, b_{-1})] - \mathbb{E}[u_1(v_1, b_{-1})] \leq \max\left(0, \mathbb{E}[b_{(2)}(v_1) \widehat{\rho}_{(2)}(v_1)] - \mathbb{E}[b_{(2)}(b_1) \widehat{\rho}_{(2)}(b_1)] + O\left(\frac{n}{T}\right)\right)$$

We give the proofs of both Theorem 4.1 and 4.2 in Section 5.3 of the appendix.

Both expectations $\mathbb{E}[b_{(2)}(v_1) \widehat{\rho}_{(2)}(v_1)]$ and $\mathbb{E}[b_{(2)}(b_1) \widehat{\rho}_{(2)}(b_1)]$ can be larger than $b_2 \rho_2$ because the $\mathbb{E}[\max_{k \neq 1} b_k \widehat{\rho}_k(v_1)] \geq \max_{k \neq 1} b_k \mathbb{E}[\widehat{\rho}_k(v_1)]$.

**Remark 4.1.** In the special case of only two advertisers, it must be that $(2) = 2$ and therefore $\mathbb{E}[b_{(2)}(v_1) \widehat{\rho}_{(2)}(v_1)] = b_2 \rho_2$ and $\mathbb{E}[b_{(2)}(v_1) \widehat{\rho}_{(2)}(v_1)] = b_2 \rho_2$. The player-regret is then very small: $\sup_{b_1} \mathbb{E}[u_1(b_1, b_2)] - \mathbb{E}[u_1(v_1, b_2)] \leq O\left(\frac{n}{T}\right)$.

The incentive can be much larger when there are more than 2 advertisers. Intuitively, this is because the bias $\mathbb{E}[b_{(2)}(b_1) \widehat{\rho}_{(2)}(b_1)] - b_2 \rho_2$ increases when $T_2(b_1), ..., T_n(b_1)$ are low–that is, it increases when the variance of $\widehat{\rho}_k(b_1)$'s are high. An omniscient advertiser 1, with the belief that $v_1 \rho_1 >> b_2 \rho_2$, can thus increase his/her utility by underbidding to manipulate the learning algorithm to allocate more rounds to advertisers $2, .., n$ and reduce the variance of $\widehat{\rho}_k(b_1)$'s. Such a strategy will give advertiser 1 negative utility in the learning CTR learning stage, but it will yield positive utility in the longer SP auction stage and thus give an overall increase to the player utility.

In the case of uniform learning, the advertiser's manipulation is limited because the learning algorithm is not significantly affected by the bid.

**Corollary 4.1.** *Let the competing bid vector $b_{-1}$ be fixed. Suppose that $v_1 \rho_1 - b_{k^*} \rho_{k^*} \geq \omega\left(\sqrt{\frac{n}{T} \log T}\right)$. If uniform learning is used in the first stage, we have that*

$$\sup_{b_1} \mathbb{E}[u_1(b_1, b_{-1})] - \mathbb{E}[u_1(v_1, b_{-1})] \leq O\left(\sqrt{\frac{n}{T} \log T}\right)$$

Nevertheless, by contrasting this with $\sqrt{\frac{n}{T} \log T}$ bound with the $\frac{n}{T}$ bound we would get in the two advertiser case, we see the negative impact of ELM bias on incentive compatibility. The negative effect is even more pronounced in the case of adaptive learning. Advertiser 1 can increase its own utility by bidding some $b_1$ smaller than $v_1$ but still large enough to ensure that $b_1 \widehat{\rho}_1(b_1)$ still be ranked the highest at the end of the learning stage. We explain this intuition with more details in the following example, which we also simulate in Section 5.

**Example 4.1.** Suppose we have $n$ advertisers and $b_2 \rho_2 = b_3 \rho_3 = ... b_n \rho_n$. Suppose that $v_1 \rho_1 >> b_2 \rho_2$ and we will show that advertiser 1 has the incentive to underbid.

Let $\Delta_k(b_1) \equiv b_1 \rho_1 - b_k \rho_k$, then $\Delta_k(b_1)$'s are the same for all $k$ and $\Delta_k(v_1) >> 0$ by previous supposition. Suppose advertiser 1 bids $b_1 < v_1$ but where $\Delta_k(b_1) >> 0$ still. We assume that $T_k(b_1) = \Theta\left(\frac{\log T}{\Delta_k(b_1)^2}\right)$ for all $k = 2, ..., n$, which must hold for large $T$ by definition of adaptive learning.

From Lemma 5.4 in the appendix, we know that

$$\mathbb{E}[b_{(2)}(b_1)\widehat{\rho}_{(2)}(b_1)] - b_2 \rho_2 \leq \sqrt{\frac{\log(n-1)}{T_k}} \leq \sqrt{\frac{\log(n-1)}{\log T}}(b_1 \rho_1 - b_k \rho_k) \quad (4.1)$$

The Eqn. (4.1) is an upper bound but numerical experiments easily show that $\mathbb{E}[b_{(2)}(b_1)\widehat{\rho}_{(2)}(b_1)]$ is in fact on the same order as the RHS of Eqn. (4.1).

From Eqn. (4.1), we derive that, for any $b_1$ such that $b_1 \rho_1 - b_2 \rho_2 \geq \omega\left(\sqrt{\frac{n}{T} \log T}\right)$:

$$\mathbb{E}[u_1(b_1, b_{-1})] - \mathbb{E}[u_1(v_1, b_{-1})] \leq O\left(\sqrt{\frac{\log(n-1)}{\log T}}(v_1 \rho_1 - b \rho_1)\right)$$

Thus, we cannot guarantee that the mechanism is approximately truthful. The bound decreases with $T$ at a very slow logarithmic rate because with adaptive algorithm, a longer learning period $T$ might not reduce the variances of many of the estimators $\widehat{\rho}_k$'s.

We would like to at this point briefly compare our results with that of [9], which shows, under an imperfect information definition of utility, that advertisers have an incentive to overbid so that the their CTRs can be better learned by the search engine. Our results are not contradictory since we show that only the leading advertiser have an incentive to underbid.

## 4.2 Bias Reduction in Estimation of the Largest Mean

The previous analysis shows that the incentive-incompatibility issue in the case of adaptive learning is caused by the fact that the estimator $b_{(2)}\widehat{\rho}_{(2)} = \max_{k \neq 1} b_2 \widehat{\rho}_2$ is upward biased. $\mathbb{E}[b_{(2)}\widehat{\rho}_{(2)}]$ is much larger than $b_2 \rho_2$ in general even if the individual estimators $\widehat{\rho}_k$'s are unbiased. We can abstract out the game theoretic setting and distill a problem known in the statistics literature as "Estimation of the Largest Mean" (ELM): given $N$ probabilities $\{\rho_k\}_{k=1,...,N}$, find an estimator $\widehat{\rho}_{max}$ such that $\mathbb{E}[\widehat{\rho}_{max}] = \max_k \rho_k$. Unfortunately, as proved by [4] and [3], unbiased estimator for the largest mean does not exist for many common distributions including the Gaussian, Binomial, and Beta; we thus survey some methods for reducing the bias.

[3] studies techniques that explicitly estimate and then substract the bias. Their method, though interesting, is specific to the case of selecting the larger mean among only two distributions. [1]

proposes a different approach based on data-splitting. We randomly partition the data in the click-through history into two sets $S, E$ and get two estimators $\widehat{\rho}_k^S$, $\widehat{\rho}_k^E$. We then use $\widehat{\rho}_k^S$ for selection and output a weighted average $\lambda\widehat{\rho}_k^S + (1 - \lambda)\widehat{\rho}_k^E$. We cannot use only $\widehat{\rho}_k^E$ for estimating the value because, without conditioning on a specific selection, it is downwardly biased. We unfortunately know of no principled way to choose $\lambda$. We implement this scheme with $\lambda = 0.5$ and show in simulation studies in Section 5 that it is effective.

## 5   Simulations

We simulate our two stage framework for various values of $T$. Figures 1a and 1b show the effect of sample selection debiasing (see Section 3, 3.2) on the expected revenue where one uses adaptive learning. (the UCB algorithm 2.1 in our experiment) One can see that selection bias harms the revenue but the debiasing method described in Section 3.2, even though it holds out half of the click data, significantly lowers the expected revenue loss, as theoretically shown in Corollary 3.1. We choose the tuning parameter $\gamma = 1$. Figure 1c shows that when there are a large number of poor quality ads, low regret adaptive algorithms indeed achieve better revenue in much fewer rounds of learning. Figure 1d show the effect of estimation-of-the-largest-mean (ELM) bias on the utility gain of the advertiser. We simulate the setting of Example 4.1 and we see that without ELM debiasing, the advertiser can noticeably increase utility by underbidding. We implement the ELM debiasing technique described in Section 4.2; it does not completely address the problem since it does not completely reduce the bias (such a task has been proven impossible), but it does ameliorate the problem–the increase in utility from underbidding has decreased.

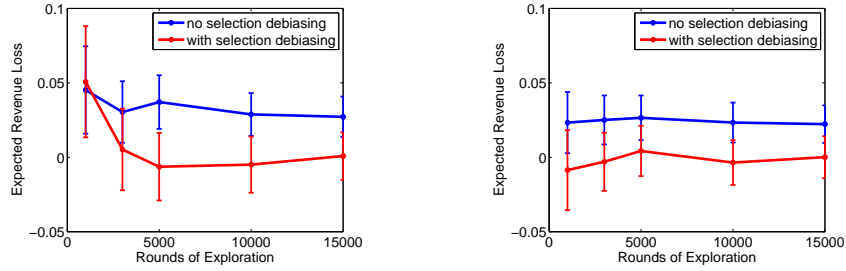

(a) $n = 2$, $\rho_1 = 0.09$, $\rho_2 = 0.1$, $b_1 = 2$, $b_2 = 1$   (b) $n = 2$, $\rho_1 = .3$, $\rho_2 = 0.1$, $b_1 = 0.7$, $b_2 = 1$

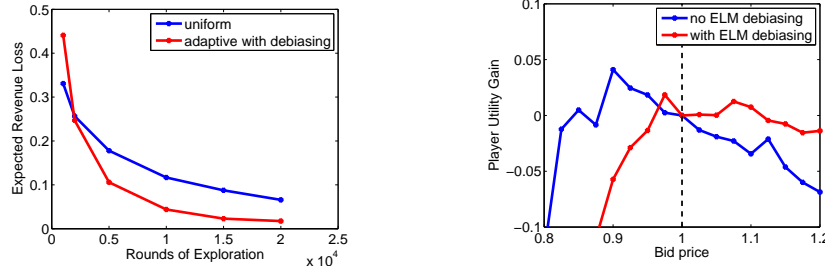

(c) $n = 42$, $\rho_1 = .2$, $\rho_2 = 0.15$, $b_1 = 0.8$, $b_2 = 1$. All other $b_k = 1$, $\rho_k = 0.01$.

(d) $n = 5$, $\vec{\rho} = \{0.15, 0.11, 0.1, 0.05, 01\}$, $\vec{b}_{-1} = \{0.9, 1, 2, 1\}$

Figure 1: Simulation studies demonstrating effect of sample selection debiasing and ELM debiasing. The revenue loss in figures a to c is relative and is measured by $1 - \frac{\text{revenue}}{b_2\rho_2}$; negative loss indicate revenue improvement over oracle SP. Figure d shows advertiser 1's utility gain as a function of possible bids. The vertical dotted black line denote the advertiser's true value at $v = 1$. Utility gain is relative and defined as $\frac{\text{utility}(b)}{\text{utility}(v)} - 1$; higher utility gain implies that advertiser 1 can benefit more from strategic bidding. The expected value is computed over 500 simulated trials.

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
