[Reviews · NeurIPS 2013]

Submitted by Assigned_Reviewer_3

This paper presents simple bias-correction for MAB algorithms for dealing with exploration and exploitation in search advertising. The method for debiasing the sample selection bias in estimating CTRs is simple, and an elegant solution to an important problem.

Some minor and major comments:
- The paper is about search advertising, but line 062 says "Each time an user visits a webpage, which we call an impression". This would be case for display advertising; Impressions occur whenever an user searches something, in search advertising.

- The paper should have additional references and discussion to literature that looks at the value of exploration in search advertising. Suggested citation:
Li, Mahdian, McAfee, "Value of Learning in Sponsored Search Auctions", Proceedings of the 6th international Workshop on Internet and Network Economics (WINE), 2010.

- In search advertising, CTRs crucially depend on context: the contextual bandit learning problem. How effectively does the bias-correction work, and how does expected revenue loss increase, with high-dimensional feature spaces?
Summary: This paper presents a simple and elegant bias-correction method for MAB algorithms effective exploration and exploitation in search advertising. I'd recommend the paper for publication if the authors revise with a relevant discussion.

Submitted by Assigned_Reviewer_5

This paper explores the task of search advertising auctions. In a second-price auction, the auctioneer must consider both the per-click price offered by the advertisers (which is known) and the probability that users will click the ad (which must be estimated from data). Multi-armed bandit algorithms can be used to choose which ads to run when estimating the click probability for each ad, since it is critical to have accurate estimates of both the highest and second-highest value advertisers in order to run the auction.

The authors identify and address two sources of bias that affect this task, impacting both the auctioneer and advertisers' profits. First, they find that multi-armed bandit algorithms tend to underestimate the value being estimated due to sample selection bias, as overestimated values are resampled (and corrected) while underestimated values receive fewer samples (and remain underestimates). In a second-price auction, the auctioneer then charges a lower price than they could on expectation if the true click probabilities were known. Bounds on this loss are proven, and the auctioneer's loss does not converge to zero due to the sample selection bias. They give an example of a simple method for correcting this bias. Second, they find that advertisers in this auction are not acting optimally by bidding their true value, as is sought with a second-price auction; instead, they can increase their expected value by under-bidding while still bidding enough that their ads are bought. The auctioneer will tend to overestimate the second price used for the auction, even if its estimates for each individual advertiser are unbiased. By underbidding an agent can cause the auctioneer to obtain better estimates of the other advertisers' rates, thus lowering the price that the winner will be charged. While this bias cannot be provably corrected for, the authors show a technique for reducing it. Empirical results are presented that demonstrate both sources of bias, and how their correction measures decrease auctioneer loss and incentivize advertisers to bid closer to their true value.

QUALITY:
The paper makes a good contribution. It demonstrates sources of bias in the search advertising task, proves bounds on them, presents corrections or reductions, and demonstrates the effects empirically.

CLARITY:
The paper is well written and easy to follow.

ORIGINALITY:
I'm far enough from this domain that it's difficult to judge originality. The sources of bias highlighted in the paper appear novel, as do the bounds and solutions.

SIGNIFICANCE:
The paper demonstrates two sources of bias in the search advertising task, corrects them, and demonstrates the resulting improvement in utility for the auctioneer. This should be a useful result to consider in future work in this domain.
Summary: The paper demonstrates the need for specialized mechanism design work in the search advertising domain, highlights new problems in current auction design, and empirically demonstrates solutions to them. This looks like a good paper, and I recommend acceptance.

Submitted by Assigned_Reviewer_6

The paper shows the naive MAB algorithms for search ads selection will produce sample selection bias, then a bias correction method is proposed to overcome the problem. The paper is well written and has strong theoretical support.

There are a few short comes of the paper:
1. This paper only analyze one ad slot situation. The multiple dynamic slot scenario is more realistic. Although the authors want to leave the multiple slot scenario to future, I am not sure if some results obtained in this paper will still hold for the realistic case.
2. The experimental section is very limited.
Summary: The paper shows the naive MAB algorithms for search ads selection will produce sample selection bias, then a bias correction method is proposed to overcome the problem. The paper is well written and has strong theoretical support. However, it lacks through experiments to support its claim.
Author Feedback

Author rebuttal: We thank all the reviewers for their helpful and encouraging comments.

Reviewer #3:
-Yes, "visits a webpage" is not accurate. It should be ``visit a search engine and submit a queryā€¯. Thanks for the catch.

-Thanks for pointing out this reference. The setting in that paper seems somewhat different as it uses $v \hat{\rho} - \E[ payment ]$ to compute player utility instead of $v \rho - \E[ payment]$. It is certainly relevant however and we will add a reference for the final draft.

-Contextual bandits are important and we are very interested in analyzing them in our future work. Sample selection bias still exists there because the number of impressions for each ad is still correlated with the percentage of observed clicks for the ad. Intuitively, we believe that the debiasing technique should still apply. As for the relationship between expected revenue loss and high-dimensional feature space, that is interesting and not clear to us. Context-dependent CTR could potentially lead to better context-dependent ranking and thus ameliorate revenue loss.

Reviewer #5:
We thank the reviewer for the detailed and helpful comments.

Reviewer #6:
-The multi-slot setting is certainly more relevant and important for PC search advertising. Having multiple slots introduces various complications, e.g. we'd need a more complex probabilistic model and we'd need to analyze non-truthful equilibria of the generalized second price auction. These complications obscure the intuition behind the two biases and we thus will tackle the multi-slot setting as an extension. The single-slot setting itself is not purely theoretical however, sponsored searches on mobile devices often have only a single slot because of the limited screen size.

-We use the experiments to corroborate and sanity check our statistical analysis. More comprehensive experiments, especially with real system and data, are certainly important and we leave that as future work.